# Extended Lymphadenectomy for Gastric Cancer in the Neoadjuvant Era: Current Status, Clinical Implications and Contentious Issues

Luigi Marano, Ludovico Carbone *, Gianmario Edoardo Poto, Valeria Restaino, Stefania Angela Piccioni, Luigi Verre, Franco Roviello and Daniele Marrelli

Unit of Surgical Oncology, Department of Medicine Surgery and Neurosciences, University of Siena, 53100 Siena, Italy
* Correspondence: ludovicocarbone1@gmail.com

**Abstract:** Despite its decreasing incidence, gastric cancer remains an important global healthcare problem due to its overall high prevalence and high mortality rate. Since the MAGIC and FNLCC/FFCD trials, the neoadjuvant chemotherapy has been recommended throughout Europe in gastric cancer. Potential benefits of preoperative treatments include a higher rate of R0 resection achieved by downstaging the primary tumor, a likely effect on micrometastases and isolated tumor cells in the lymph nodes, and, as a result, improved cancer-related survival. Nevertheless, distortion of anatomical planes of dissection, interstitial fibrosis, and sclerotic tissue changes may increase surgical difficulty. The collection of at least twenty-five lymph nodes after neoadjuvant therapy would seem to ensure removal of undetectable node metastasis and reduce the likelihood of locoregional recurrence. It is not what you take but what you leave behind that defines survival. Therefore, para-aortic lymph node dissection is safe and effective after neoadjuvant chemotherapy, in both therapeutic and prophylactic settings. In this review, the efficacy of adequate lymph node dissection, also in a neoadjuvant setting, has been investigated in the key studies conducted to date on the topic.

**Keywords:** lymphadenectomy; gastric cancer; neoadjuvant

## 1. Current Status and Clinical Significance of Lymph Node Dissection

### 1.1. D2 Lymphadenectomy

The lymphatic route is the main way that gastric cancer spreads; therefore, lymph node involvement represents the leading prognostic factor after surgery [1,2].

D2 lymphadenectomy currently represents the standard procedure for locally advanced gastric cancer [3,4]. According to the third and fourth edition of the Japanese guidelines, D2 lymphadenectomy for subtotal gastrectomy defines the surgical removal of stations 1, 3, 4sb, 4d, 5, 6, 7, 8a, 9, 11p, and 12a, while D2 lymphadenectomy for total gastrectomy includes stations 1–7, 8a, 9, 11p, 11d, 10, and 12a [5,6]. In the fifth Japanese guidelines [3], station 10 was excluded in D2 dissection for total gastrectomy, except for tumors located in the upper third of the stomach along the greater curvature. Current guidelines indicate D2 lymphadenectomy in all cN+ or cT2-4 tumors, while indicating D1 dissection (limited to stations 1–6) for cT1a tumors or for differentiated cT1b smaller than 1.5 cm in diameter [4,7,8].

The optimal extent of lymphadenectomy has been a topic of investigation in the past decades. Adequate lymphadenectomy resulted in four potential clinical benefits: (1) adequate disease staging due to the increased number of nodes retrieved; (2) the removal of potentially metastatic lymph nodes, resulting in increased surgical radicality; (3) a decrease in GC recurrence, especially locoregional; (4) prognostic benefit by potential improvement in long-term survival [9].

### 1.1.1. Adequate Disease Staging Due to the Increased Number of Nodes Retrieved

The radicality of the surgical procedure can be evaluated by the number of harvested lymph nodes. When the number of retrieved nodes is less than 10, it is extremely unlikely that we can adequately determine the truly negative incidence of regional lymph nodes [10–12]. For patients with T1-2 N0 stage, the risk of downstaging with an inadequate number of removed lymph nodes is especially high, therefore underestimating the need for a more radical lymphadenectomy [12–16]. Although the American Joint Committee on Cancer (AJCC) does not define a standard threshold for the number of nodes to be considered adequate in assigning an appropriate N stage, 16 has been recommended as the minimum number for adequate lymphadenectomy by the Japanese, Korean and European guidelines [12,17–21], while 15 has been suggested as the minimum number by British and North American societies [12,22–24]. The Italian Research Group for Gastric Cancer (GIRCG), based on data collected from a prospective database of 2822 patients treated in high-volume centers (78% and 53% of cases with more than 15 and 25 removed nodes, respectively), highlighted that overall postoperative mortality was 3.5%, even when including aged patients or advanced stages [9,25]. Additionally, an analysis of 40,281 patients, included in the American National Cancer Database for gastric cancer, showed a higher 30-day mortality after >29 nodes were removed (4.3%) compared to a resection of 15–28 nodes and <15 nodes (3.0% and 2.1%, respectively) [26,27]. In a recent update from the GIRCG database, which included 6230 patients, a strong correlation between the number of removed and metastatic lymph nodes was observed, with a plateau above 40–50 lymph nodes examined (Figure 1).

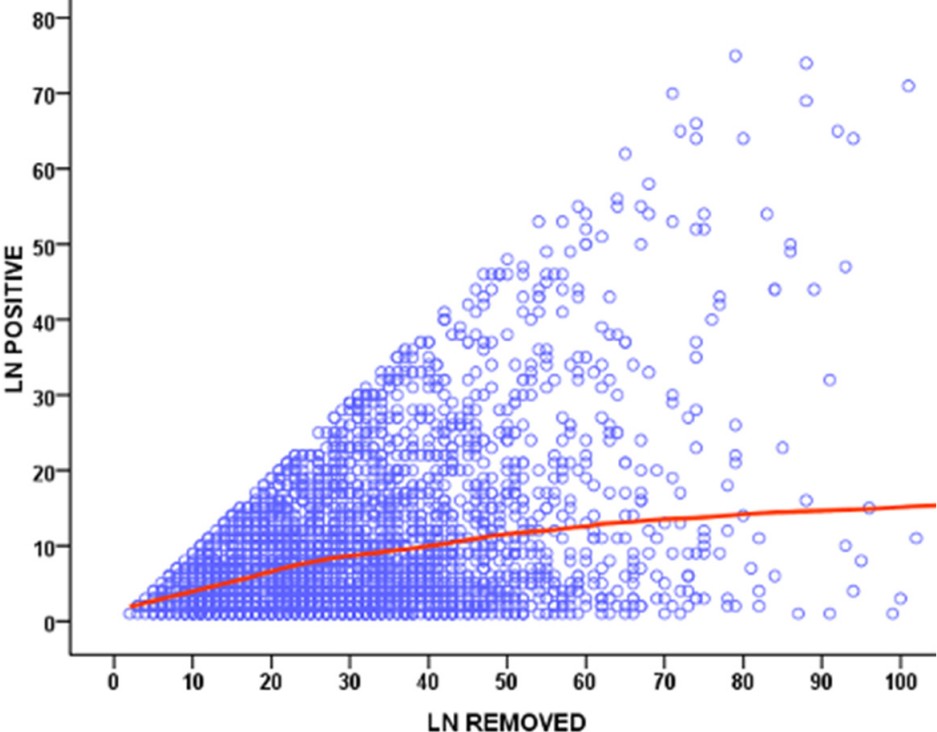

**Figure 1.** Scatter plot showing the correlation between the number of removed and metastatic lymph nodes (LN) in GIRCG cohort.

### 1.1.2. Removal of Potentially Metastatic Lymph Nodes, Resulting in Increased Surgical Radicality

Removing the involved nodes could improve surgical radicality. The number of positive nodes is a well-established prognostic factor in gastric cancer (Figure 2). However, the GIRCG experience, similarly to other reports [28,29], indicates that the survival probability in patients with the same number of involved nodes is strongly affected by the number

of removed nodes (Figure 3). For instance, the survival probability of patients with three involved nodes is 12% when less than 10 nodes are removed, 26% when 11–15 nodes are removed, and 60% when more than 25 nodes are examined. Similar behavior is observed for other lymph-node positive subgroups, with a plateau above 45 removed lymph nodes. These survival trends may be due to the stage migration, the "Will Rogers phenomenon", but we strongly believe that the therapeutic effect of extended lymphadenectomy plays a crucial role in prognostic benefit. Furthermore, the removal of potential micrometastasis, even in node-negative patients, could also improve the oncological outcome [30,31]. In surgical practice, not only the total number of removed lymph nodes is important, but also the topographic sites of different nodal stations, according to the Japanese Gastric Cancer Association (JGCA) guidelines [32,33]. The term "contamination" was used to define over-treatment in a certain group (i.e., when a surgeon dissected two or more lymph node stations which he should not have), while the term "non-compliance" was used to define under-treatment (i.e., when a surgeon did not dissect two or more lymph node stations which he should have). The D2 non-compliance mainly involved nodal stations 10, 11d, and 12a in total gastrectomy and 4sb, 11p, and 12a in subtotal gastrectomy. Contamination mainly consisted of the removal of posterior stations (8p, 12b, 12p) in total gastrectomy and 8p, 11d, 12b, and 12p in subtotal gastrectomy. In previous literature, "standard" lymphadenectomy is usually referred to as the surgical dissection described by the JGCA, while "extended", "more extended" and "super-extended" refer to D2, D2 plus and D3 dissections, respectively. As mentioned above, the D2 lymphadenectomy is routinely accepted in most international guidelines [4]. Despite a more extended procedure (D2plus) gaining acceptance among surgeons, D3 seems to fail in obscurity in recent years. In our opinion, more extended lymphadenectomy should be performed in selected cases at risk of metastasis to posterior or para-aortic lymph nodes (PAN), while proximal tumors or diffuse-type tumors are particularly prone to spread to distant nodes and may benefit from both posterior and PAN dissection [34].

1.1.3. Decrease in GC Recurrence, Especially Locoregional

The increase in R0 obtained with extended lymphadenectomy has been associated with a decrease in locoregional tumor recurrence [9].

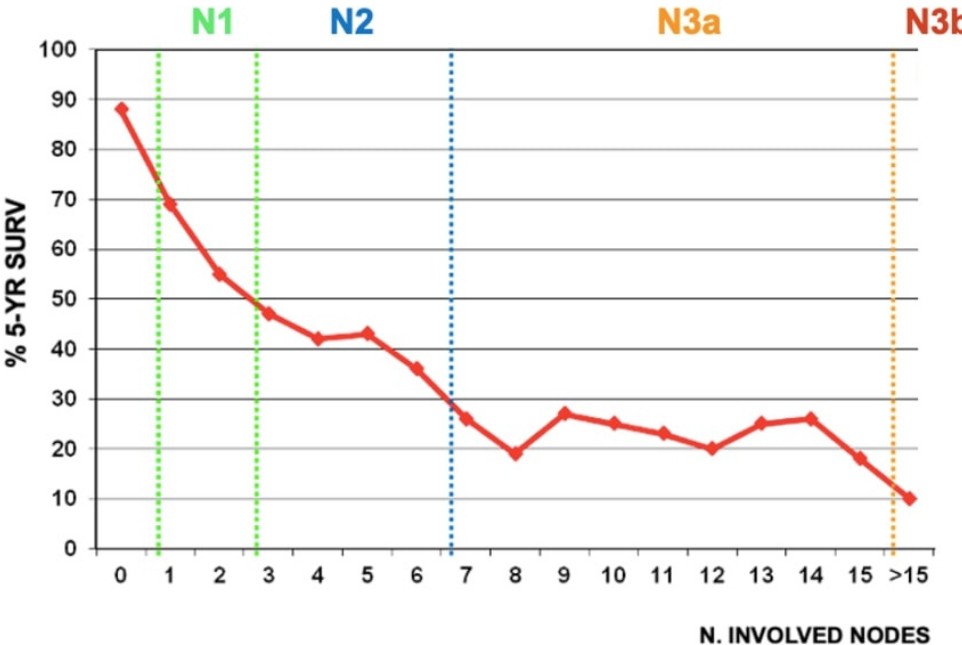

**Figure 2.** Correlation between the number of metastatic nodes and 5-year overall survival in GIRCG cohort. "N" classification according to the 8th Edition of the AJCC is showed in subgroups.

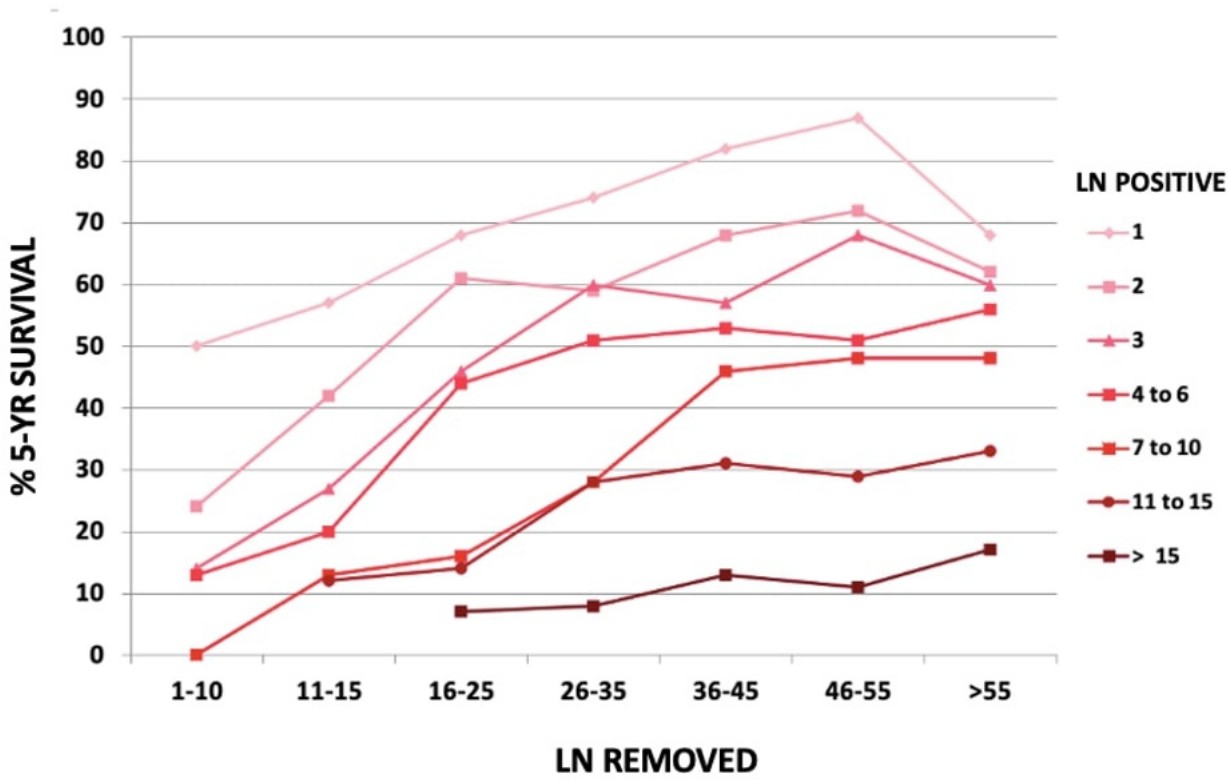

**Figure 3.** Five-year overall survival according to the number of removed nodes in different node positive subgroups, in GIRCG cohort.

1.1.4. Prognostic Benefit by Potential Improvement in Long-Term Survival

The Dutch trial clearly observed lower loco-regional recurrences, as well as cancer-related mortality in the D2 arm compared to the D1 arm, when considering only long-term mortality without 30-day postoperative mortality [35,36]. An Italian trial noted a trend toward statistical significance in survival benefit, when compared lymph node-positive patients who underwent D2 and D1 gastrectomy [37–40]. The abovementioned evidence is confirmed in a Taiwanese randomized trial, which compared limited and extended lymph node dissection [41]. On the other hand, the surgical removal of the spleen and the pancreatic tail to achieve stations 10 and 11 strongly affects morbidity and mortality rates [42].

Key points:
- D2 lymphadenectomy is the standard surgical treatment with curative intent for advanced gastric cancer.
- Adequate D2 enables accurate disease staging, reduces the incidence of locoregional recurrences and contributes to an improved long-term survival.
- Undetectable node metastases are associated with high rates of locoregional recurrence.

*1.2. D2 plus and D3 Lymphadenectomies*

Radical surgery results from the balance of two elements: the need to guarantee optimal locoregional control of the disease, and the clinical cost of extensive surgery.

The extent of lymphadenectomy, defined by the amount of lymph nodes and the quality of the harvested stations, is a marker of quality of surgical resection. A Turkish anatomical study, comparing total gastrectomy with extended lymphadenectomy to station 16 on autopsy and in vivo cases, revealed higher numbers of lymph nodes harvested in autopsy cases at stations 10, 11, 12, and 16, but no difference at stations 1, 2, 3, 6, 7, 8, and 9 [43].

D2-plus lymphadenectomy is indicated by Japanese studies only in specific cases [3,6]: (i) D2+ station 14v consisting of dissection of superior mesenteric venous lymph nodes for cancer of the distal stomach with metastasis to the station 6 lymph nodes, (ii) D2+ station 13 with dissection of lymph nodes posterior to the pancreas head for cancer invading the duodenum, and (iii) D2+ station 16 with dissection of abdominal aortic lymph nodes for cancer with extensive lymph node involvement.

Historically, the so-called D3 dissection included both PAN (16a2 and 16b1) and "posterior" stations (8p, 12p and 13) [44]. However, as since the third edition of the Japanese guidelines, routine removal of lymph node stations beyond the standard D2 is no longer indicated; more extended procedures beyond the D2 are now called D2plus.

In a recent trial, the survival analysis indicated similar 3-year outcomes between D2 and D2+ arms, while, among patients with duodenum involvement, the 3-year disease-free survival rate of the D2+ group was higher than the D2 group (61.5% vs. 20%, respectively) [45].

A closer look at the literature on the definition of "extensive lymph node metastases (ELM)", however, reveals a number of gaps and shortcomings. In most cases, PAN enlargement without other lateral metastasis is considered ELM. From an anatomical point of view, this definition includes the stations 16a2-b1 (lower end of celiac trunk and upper border of inferior mesenteric artery); in a preoperative radiological examination, a positive (metastatic) PAN found that the longest axis of the lymph node is at least 10 mm. As we consider the latest RECIST (Response Evaluation Criteria in Solid Tumors), a cut-off of ≥15 mm in the short axis should be analyzed [46]. Our experience based on analysis of CT scans found that ≥8 mm in the short axis is the value that we should take as suspected of PAN gastric cancer metastases [47]. The other definition of ELM was proposed by Japan Clinical Oncology Group (JCOG) studies about neoadjuvant treatment with ELM. They proposed to include not only PAN but also bulky nodes (bulky N2), defined as lymphatic tissue with a diameter of ≥3 cm or at least two adjacent nodes of ≥1.5 cm in diameter surrounding the coeliac artery and its branches [48–50]. (Figure 4) The reason for such a consideration is the inability to perform adequate resection, whereas even in the case of resection, the prognosis remains poor, similar to PAN metastases. In the JCOG0405 trial, survival outcomes were similar in PAN-only metastases as well as bulky N-only, while the worst outcome was for patients presenting metastases in both bulky N2 and PAN (results will be discussed further below) [48].

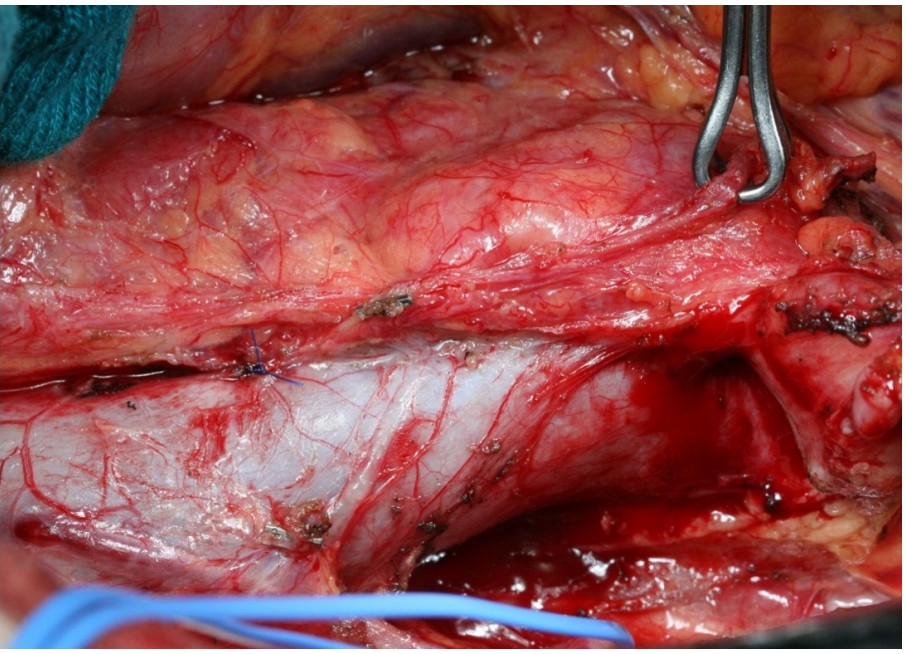

**Figure 4.** Bulky nodes in 16a2-b1 stations.

In 2006, in fact, the phase III JCOG9501 randomized controlled trial investigated the role of "prophylactic" D2+ para-aortic lymph node dissection (PAND) compared to D2 in locally advanced gastric tumors without clinically evident metastases in the para-aortic area [51]. (Figure 5) Prophylactic D2+ PAND did not result in survival improvement, while it was associated with a higher morbidity rate. Therefore, the prophylactic removal of para-aortic lymph node 16a2 and 16b1 stations may not be recommended in all patients. Moreover, "curative" D2+ PAND, i.e., dissection of the para-aortic area in cases of clinically detected lymph nodes, was discouraged due to the rate of poor survival [52].

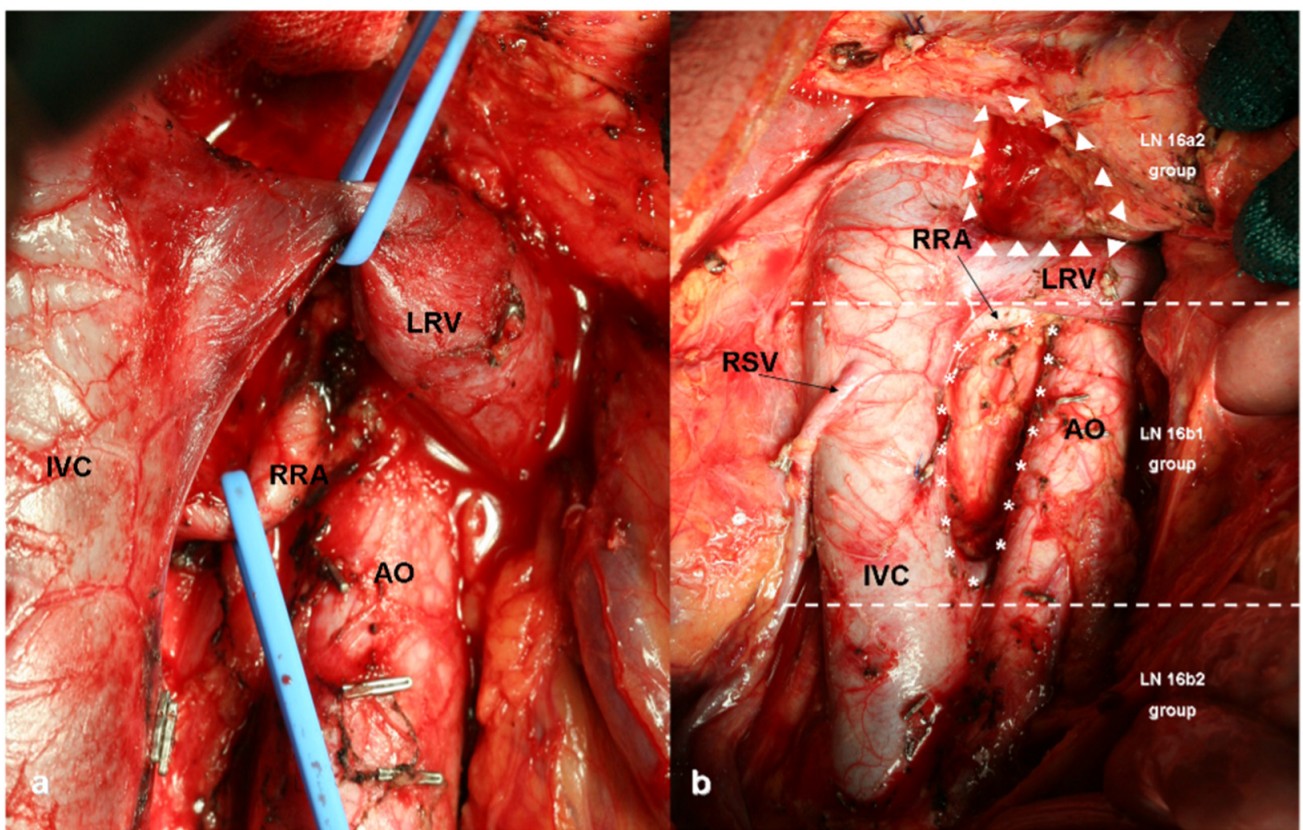

**Figure 5.** Para-Aortic Lymph Node Dissection (PAND): (**a**) intraoperative view of 16a2 lymph node dissection; (**b**) 16a2 (upper third), 16b1 (medium third) and 16b2 (lower third) lymph nodes stations. IVC: Inferior Vena Cava, RRA: Right Renal Artery, LRV: Left Renal Vein, AO: Aorta, RSV: Right Spermatic Vein, LN: Lymph Nodes.

The stations 13 (retropancreatic) and 14v (superior mesenteric) are commonly considered the "gatekeepers" of the para-aortic nodes, and their dissection in middle- and lower-third advanced gastric cancer represents an independent prognostic factor when performed by experienced surgeons. [53–55].

The issue of the other "posterior" stations such as 8p and 12p has not been deeply discussed. Our group recently explored the benefit of "posterior" station removal in a large Western cohort undergoing D2plus dissection [56]. Posterior stations are "regional" nodes in distal tumor locations and expressions of an extensive nodal spread. Interestingly, middle- and lower-third gastric tumors showed a higher trend to metastasize to posterior nodes than the upper-third, whereas for PAN metastases it was the reverse. Only 6.3% of patients had metastases in posterior stations, showing a 5-year survival rate of 17% after adequate lymphadenectomy. Moreover, some retrospective data on D2plus lymphadenectomy showed that a higher probability of long-term survival was achieved in pT3 and pT4a patients with PAN metastasis [34,57].

Recent studies from the GIRCG, comparing D2 with D2plus lymphadenectomy in patients who underwent upfront surgery with both prophylactic and curative intents, showed that D2plus offers a better locoregional control in diffuse histotype [58]. Tumor histology largely affects the surgeon's choice concerning the extent of lymphadenectomy. A European survey concluded that more extended lymphadenectomy was selected for tumors with Lauren diffuse histotype, even if no nodal metastases were detected on preoperative workup [59].

More extended lymphadenectomy should only be performed in dedicated high-volume hospitals, due to the high risk of post-operative complications and mortality [60].

## 2. Evolution in the Surgical Management of Gastric Cancer

It is not easy to offer a tailored lymphadenectomy (including sentinel node navigation surgery) for specific subgroups of patients, since the lymphatic network of the stomach is very complex. Specifically, the incidence of positive lymph node is related to the depth of invasion, and metastasis rates range from 2.3% for mucosal invasion to 86.6% for serosal pathological invasion. Nonetheless, most positive nodes are microscopic metastases, which often occur in small-sized lymph nodes, resulting in a difficult clinical and intraoperative diagnosis. Thus, gastrectomy with prophylactic lymphadenectomy is recommended as the standard surgical procedure for gastric cancer [61].

Widened lymphadenectomy involves mobilization of the duodenum, which is obtained performing the Kocher maneuver. This access provides exposure of the para-aortic node stations and improves the surgical exposure of lymph node stations 8p, 12p, and 13. Nodal stations 16a2 and 16b1 are included in PAND for gastric cancer. PAND is the lymphadenectomy performed between the upper margin of the origin of the celiac artery and the lower border of the left renal vein (station 16a2), and the lymph nodes between the lower border of the left renal vein and the upper border of the origin of the inferior mesenteric artery (station 16b1) [62].

Despite radical surgery still being the main treatment modality in gastric cancer, the validity of the complete mesogastric excision is still debated because of embryological restrictions of the gastric mesentery. Systematic mesogastric excision is a procedure consisting of three main steps: (i) the mesogastrium is dissociated from the adjacent mesenteries or the parietal wall following the embryological planes; (ii) the fat tissue containing lymph nodes is separated from the pancreas and its associated vessels; and (iii) the tumor-specific mesenteries are dissected. Therefore, D2 gastrectomy may be defined as the "en bloc" resection of the primary tumor and the tumor-specific mesenteries [63].

Interestingly, several studies have applied near-infrared (NIR) fluorescence imaging to gastric cancer for performing surgery in a real-time manner, including localizing cancers, assessing surgical margins, tracing lymph nodes, and mapping critical anatomical structures [64]. With the aid of lymphography, a significantly higher number of lymph nodes may be harvested [65]. However, some clinical situations, such as linitis plastica, post-chemotherapy fibrosis reaction, or cytotoxicity, and disturbed lymphatic flow following preoperative endoscopic submucosal dissection in early gastric cancer, could limit NIR application. From a pathological point of view, tumor cell metabolism is altered by chemotherapy drugs which lead to several histopathological changes, such as eosinophilic appearance of the cytoplasm, nuclear enlargement, condensation, cell necrosis, cell dissection, and destruction of ducts, neutrophils, and other inflammatory cells. Tumor cells are eventually replaced by the infiltration and aggregation of monocytes which cause interstitial fibrosis, which then leads to a shrinkage of the primary tumor and metastatic lymph nodes. This phenomenon does not allow lymphatic drainage and, therefore, may alter the visualization via indocyanine green tracer (ICG) [66]. At the same time, however, NIR has been proved to reduce intraoperative blood loss without increasing the operation time and overall complications, and to improve lymphadenectomy in terms of total number of lymph nodes harvested when performing minimally-invasive radical gastrectomy after neoadjuvant therapy (40.8 vs. 31.8), especially in 5–12a subgroups (19.1 vs. 13.2), and

to reduce the lymph node non-compliance rate (35.1% vs. 51.1%). Clearly, the effect is marked in patients who underwent neoadjuvant therapy but with poor chemotherapy effects (23.3% vs. 55%) [66]. To our knowledge, there is no prospective research to confirm whether ICG technology can increase the number of lymph node dissections for patients who had undergone neoadjuvant chemotherapy. The study NCT04611997 explores the feasibility and the effect on 3-year disease-free survival using laparoscopic ICG tracing [67].

## 3. Contentious Issues in the Era of Multimodal Strategy

### 3.1. Preoperative Therapy

In 2012, De Manzoni and Roviello described D2 lymphadenectomy as "safe when conducted in specialized centers", and D3 as "an emerging technical approach especially indicated for patients with upper-third gastric cancer" [68]. Is this still the case in the era of neoadjuvant therapy?

For many years, surgery has been the only treatment option in patients affected by gastric cancer. After the Medical Research Council Adjuvant Gastric Infusional Chemotherapy (MAGIC) trial and the Fédération Nationale des Centres de Lutte contre le Cancer (FNCLCC) and the Fédération Francophone de Cancérologie Digestive (FFCD) trial in the Western world, the neoadjuvant approach has become a gold standard in treatment of locally advanced gastric cancer in Western countries and in parts of the Asia–Pacific region [38,69,70]. Neoadjuvant chemotherapy increases the chance for R0 resection, eliminates early micrometastases, downstages the lesions, and thus improves long-term outcomes [38], increasing the overall survival rate by 9% and R0 resection by 1.4-fold [71]. Another potential benefit may be the possibility for patients to receive systemic chemotherapy without having to wait for the surgery and the related postoperative recovery [72].

To date, FLOT schedule seems to provide the most successful results [73]; selected patients may also benefit from the use of fluoropyrimidine-platinum doublet or triplet chemotherapy [70]. The influence of human epidermal growth factor 2 (HER2)-targeted agents and vascular endothelial growth factor (VEGF) inhibition were recently analyzed in the metastatic setting. A new KEYNOTE-811 study suggested that adding pembrolizumab to standard therapy with trastuzumab and chemotherapy reduces tumor size and induces complete responses in unresectable or metastatic HER2-positive gastric cancers [74]. In the PETRARCA trial (NCT02581462), the combination of trastuzumab, pertuzumab, and perioperative FLOT showed improvement in the pathologic complete response and nodal negativity rates [75]. In the RAMSES/FLOT7 trial (NCT02661971), the addition of ramucirumab to FLOT has been showed to improve R0 resection rates but not the pathologic response rate [76,77].

The National Comprehensive Cancer Network (NCCN) [78] and the Italian Association of Medical Oncology (AIOM) [79] recommend neoadjuvant chemotherapy for resectable gastric cancer with clinical stage ≥ T2 or N+, while the European Society for Medical Oncology (ESMO) [80] indicate perioperative chemotherapy for patients with stage ≥ IB resectable gastric cancer. The GIRCG suggests considering it for clinical stage ≥ T3 and/or metastatic nodes, as the 5-year survival probability of T1 and T2 node-negative cases was greater than 80% in their series [81].

All studies conducted so far have demonstrated the superiority of preoperative chemotherapy compared to surgery alone, whether followed or not by adjuvant chemotherapy, while perioperative chemotherapy has rarely been investigated [82]. The GastroDOC trial investigated the compliance with neoadjuvant (four cycles of DOC regimen, i.e., docetaxel, oxaliplatin and capecitabine) and perioperative (two cycles pre- and two cycles post-surgery) therapy of stage II and III cancer patients, showing the neoadjuvant treatment to be more frequently completed [83].

Another critical open question is whether preoperative chemotherapy should be administered, taking into account the tumor's (1) histopathological and (2) molecular characteristics.

### 3.1.1. Histopathological Characteristics

In diffuse-type tumors, only 14% of patients had a good pathological response, while a retrospective analysis from a French National registry suggested no benefit in signet-ring-cell type gastric cancer [84]. Similarly, in an early study aiming to evaluate the prognosis of patients who underwent neoadjuvant chemotherapy followed by surgery with PAND, patients with a diffuse-type histology tended to develop a worse prognosis (8-year survival halved when compared with the intestinal subtype) [85]. From a side-study of the multicenter LOGICA trial, lymph node metastases were more frequent in diffuse than intestinal tumors (66% vs. 52%, respectively), but not in cT3-4 than cT1-2-stage (59% vs. 51%, respectively) [86].

### 3.1.2. Molecular Characteristics

Following the introduction of molecular classifications [87–89], several studies have been conducted to clarify their potential impact in clinical decision-making and treatment of gastric cancer [47,90]. A different prognosis was attributed to the microsatellite instability (MSI) showing the best survival rate, and to the epithelial-mesenchymal transition (EMT) confirming the worst prognosis. The condition of MSI reflects approximately 10% of the cases of operable gastric cancer [91,92]. Although MSI was found to be a positive prognostic factor in the post hoc analysis of the MAGIC and the CLASSIC trials, the use of perioperative or adjuvant chemotherapy was not effective or even potentially detrimental. The same results were confirmed in the post hoc analysis of the CRITICS study and the ITACA-S trial [93,94]. Similarly, in 2019 a meta-analysis further demonstrated that MSS patients could benefit from adjuvant chemotherapy [95]. However, the limited number of MSI-high patients enrolled did not allow significant conclusions to be drawn. In our experience, MSI status was associated with a high rate of N0 stage (53.7% vs. 29.7%), a lower number of lymph node metastases (1 vs. 5), a less extensive spread to lymph node stations, and no skip metastases (0 vs. 6.1%) than microsatellite stable (MSS) tumors [91]. We therefore speculate that an approach consisting of preoperative therapy and more extended lymphadenectomy may not be indicated in all MSI tumors, due to the weak propensity to spread beyond the limit of the standard D2 dissection. Moreover, clinical trials enrolling advanced MSI-high gastric cancer patients showed prolonged survival following immunotherapy (pembrolizumab monotherapy), even in the first-line setting [76]. Similarly, Epstein–Barr virus (EBV)-associated gastric cancer, influencing tumors progression in about one patient in ten [96], has been described as being more susceptible to immune-checkpoint blockade, due to its increased PD-L1 and PD-L2 expression [76]).

Key points:

- Neoadjuvant chemotherapy is indicated in clinical stage > T2 or N+.
- Preoperative treatments increase R0 resection and improve overall survival.
- In cT3-T4 diffuse histotype, perioperative chemotherapy should be considered when staging laparoscopy confirms no peritoneal metastasis and negative peritoneal lavage cytology.
- MSI tumors may not benefit from chemotherapy and more extended lymphadenectomy.

### 3.2. Lymphadenectomy after Neoadjuvant Chemotherapy

After 10 years, the same authors raise the question of "whether a certain level of variability would reflect more likely the efforts by expert surgeons to choose the best treatment for their individual patients rather than technical issues" [97].

While the literature abounds with trials on the different choices of lymphadenectomy in patients undergoing upfront surgery, there is wide variation in practice patterns for lymph node dissection after neoadjuvant therapy. Studies have proven that patients who underwent neoadjuvant chemotherapy need more lymph node dissections to truly reflect the tumor stage and prognosis. The cut-off of 25 lymph nodes seemed to have had the best predictive power in terms of morbidity and mortality [98].

The collection of a high number of lymph nodes is associated with improved survival only for node-positive patients [28,99,100]. Although "extensive lymphadenectomy" is commonly practiced by surgeons in European high-volume centers, the extent of lymphadenectomy during gastrectomy for gastric cancer in the USA and in Asia is variable. Interestingly, an early report from the National Cancer Database proved no lymph nodes in the resection specimen of a quarter of gastrectomy performed [99,101].

Surgeons are forced to perform a more adequate lymphadenectomy in a condition of chemotherapy-driven tissue sclerosis and fibrosis. Since peripheral metastatic nodes may be difficult to palpate intraoperatively, the number of lymph nodes harvested greatly varies depending on the skills of the technician. Practically, these technical difficulties led to longer operative times, inadequate lymph node sampling, and increased blood loss [102]. In our experience, the fibrosis induced by neoadjuvant chemotherapy is particularly evident in the peri-pancreatic area and the retroperitoneal area (Figure 6).

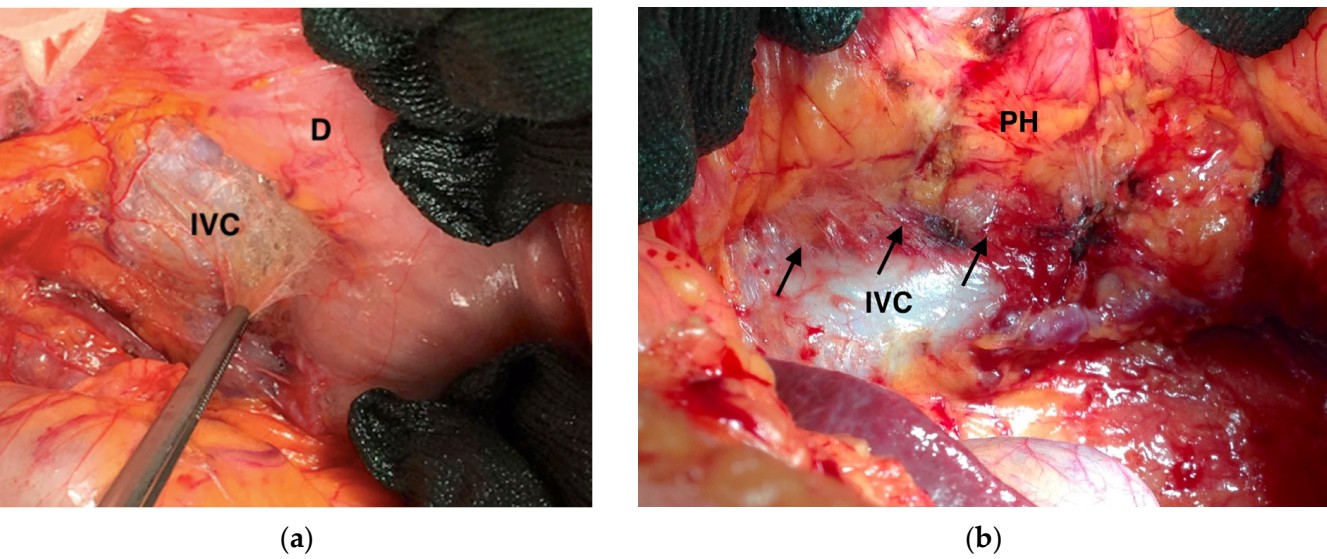

(**a**) (**b**)

**Figure 6.** Intraoperative frames highlighting fibrosis induced by neoadjuvant chemotherapy (black arrows): (**a**) Kocher's maneuver; (**b**) Kocher's maneuver showing adhesion between inferior cava vein, ovarian vein, and pancreatic head. IVC: Inferior Vena Cava; D: Duodenum; PH: Pancreatic Head.

As such, surgeons' experience and hospital quality of care could have a strong impact on D2+ dissection after pre-operative chemotherapy. Gastrectomy with D2 is a complex surgical procedure that requires a significant learning curve [12,103,104]. A large South Korean study demonstrated that node retrieval and overall survival rate of patients was improved by institutional experience, and that surgeon subspecialty was an independent predictor of improved outcomes [105]. Most studies, including that of the Italian Gastric Cancer Collaborative, suggest a learning curve of approximately 15–25 cases or between 8 and 24 months in establishing a plateau [106,107].

In 2017, the Dutch Upper Gastrointestinal Cancer Audit (DUCA) group designed the Textbook Outcome, a multidimensional measure representing an ideal course after cancer surgery [108]. It consists of ten perioperative quality-of-care parameters, including adequate lymphadenectomy with "greater than 15 lymph nodes sampled" which, if met, are associated with better survival rates in gastric cancer. This suggests that a good patient selection, well-performed surgery, and optimal postoperative care are ways to allow fast discharge, optimize long-term outcome, and lower costs for the healthcare system [109,110]. In 2021, perioperative chemotherapy was added to the other features, resulting in a new Textbook Oncological Outcome. Parallel to it, GIRCG is conducting a consensus study aimed to modulate specific parameters (i.e., compliance/adherence to chemotherapy, or 25 lymph nodes sampled in neoadjuvant setting), defining a more actual and specific definition of Textbook Outcome in Gastric Surgery (TOGS).

It is intuitive, but also clearly demonstrated in all studies dealing with this issue, that the higher number of lymph nodes required to be excised and the sequelae of chemotherapy treatments further increase the learning curve. This implies that the extended lymph node dissection after neoadjuvant treatment could be associated with an increase in intra- and post-operative complications. Shrikhande described a 12% morbidity rate and no mortality in his series, mainly treated with EOX regimen [111]. Additionally, in the same study, the median blood loss and the number of lymph nodes harvested were similar to upfront surgery (500 vs. 400 mL, and 16 vs. 18 lymph nodes), but positive nodes were noted even in patients with a major pathological response of the primary tumor. In contrast, Wu observed greater intraoperative blood loss among neoadjuvant patients compared with those who underwent upfront surgery, probably due to the abovementioned chemotherapy-induced fibrosis surrounding the lymph nodes [112]. Thus, the administration of preoperative treatment does not obviate the need for a complete radical resection and D2 or more extended lymphadenectomy, which, although involving greater technical difficulties and longer operative time, may still be achieved with suitable oncological resectability [47].

Key points:

- The collection of more than 25 lymph nodes is associated with improved survival in N+ patients who underwent neoadjuvant chemotherapy.
- Good responders to neoadjuvant chemotherapy may have nodal micrometastases.

### 3.3. Closed Trials

The Stomach Cancer Study Group of the Japan Clinical Oncology Group (SCSG/JCOG) has conducted the following trials since 2000, in order to investigate how to improve the therapeutic effects of preoperative chemotherapy followed by gastrectomy with D2 plus PAND for gastric cancer patients with ELM:

JCOG0001: Concerning extended and super-extended lymphadenectomies, the phase II clinical trial enrolled patients who had received preoperative chemotherapy according to the schema irinotecan and cisplatin, and gastrectomy with D3 dissection for gastric cancer with ELM. Despite the study being stopped due to three treatment-related deaths, the available results showed a median survival of 14.6 months, and a 3-year survival rate of 27% [49].

JCOG0405: At present, PAND is considered safe after neoadjuvant chemotherapy in patients with bulky nodes in the D2 stations with or without lymphadenopathy in the para-aortic 16 a2–b1 area. Such a statement mainly derives from the results of a phase II trial [48]: patients with clinically detected ELM (i.e., the presence of bulky nodes in the D2 stations, with or without positive PAN (16 a2–b1)) could benefit from a multidisciplinary treatment including two courses of neoadjuvant chemotherapy with S-1 and cisplatin, followed by D2 plus PAND, with a 5-year survival rate of 53%. The presence of bulky nodes at the D2 stations or positive PANs alone demonstrated 3-year survival rates of 68% and 57%, while both conditions reached just 17% [48].

Extended surgery to PAN might be a promising and effective approach in prophylactic (bulky D2 nodes) intent to improve patients' survival, due to closure of lymphatic pathways or presence of undetectable micrometastases.

JCOG1012A: In 2019, an integrated analysis of the two previous phase II trials (JCOG0001 and JCOG0405) regarding gastric cancer with ELM showed better survival for S-1 plus cisplatin scheme with respect to the irinotecan plus cisplatin scheme. Moreover, the 5-year overall survival rate was poor for patients with both positive bulky nodes and PAN (19.2%), compared with positive bulky nodes alone (50.7%) and positive PAN alone (43.5%) [113].

JCOG1002: On the basis of the significative results of a retrospective study demonstrating a 2-year overall survival rate of 93.8% [114], a phase II clinical trial using docetaxel, cisplatin and S-1 as neoadjuvant chemotherapy in advanced gastric cancer with ELM showed a response rate of 57.7%, according to the Response Evaluation Criteria in Solid Tumors (compared to 51% of the JCOG0405 study), with R0 resection achieved in 84.6% of patients (compared to 82% of the JCOG0405 study) [115].

Interestingly, Yoshida investigated the long-term survival rate of patients treated with chemotherapy alone for advanced gastric cancer. Overall, 2- and 5-year survival rates were respectively 14.3% and 10.4% for patients with metastases to abdominal nodes only [116]. Other studies obtained similar results, with a 3-year survival rate of 13.1% in the case of PAN-only metastases [117]. This evidence supports the pivotal role of surgery in the treatment of gastric cancer with curative intent.

Proper selection of patients currently remains the main issue as preoperative tumor staging is based only on radiological findings [118]. Indeed, preoperative nodal assessment is mainly obtained with the aid of computed tomography, but the specificity of this method is low because it is mainly based on the size of the lymph node [119]. In view of all the above considerations, the removal of para-aortic nodes is recommended with a "curative" intent; therefore, when para-aortic nodes register a good pathological response at restaging clinical examinations.

A Korean study analyzed a large cohort of patients who had undergone gastrectomy for gastric cancer: it very rarely extended to PAN (1.3% of cases) and described a 5-year locoregional recurrence rate of 8.5%, mostly in the stations 16a2 and 16b1 (46 and 60%, respectively) [120]. Regardless of the location of the primary tumor, indeed, lymphatic flow ultimately drains into the para-aortic region. It has been assumed that a subset of patients with advanced gastric cancer may have PAN micrometastases or metastases not detectable through the preoperative imaging examinations. A "prophylactic" PAND would prevent locoregional recurrence, especially in good responders to preoperative chemotherapy.

Another interesting aspect is that no clear benefit of neoadjuvant therapy has been found in older adults. This is because on the one hand, most of the studies conducted so far have excluded patients older than 70 years, and, on the other hand, elderly patients are more likely to die from non-cancer-related causes than younger patients [121].

Key points:

- PAND is safe and effective after neoadjuvant chemotherapy, in both therapeutic (positive PAN) and prophylactic (positive bulky nodes in D2 and negative PAN) surgery.
- Neoadjuvant chemotherapy followed by D2+ PAND improves long-term survival [122].
- Good responders to preoperative chemotherapy better benefit from PAND.

### 3.4. Benefits and Drawbacks of Minimally Invasive Approach

High referral centers for minimally invasive surgery are aiming to achieve a patient-targeted treatment [123–125]. It is supported that laparoscopic gastrectomy offers similar rates of survival when compared to open gastrectomy in both early and advanced gastric cancer [126]. Moreover, while laparoscopic D2 distal gastrectomy has been demonstrated as feasible and safe [123,127,128], laparoscopic total gastrectomy still represents a surgical challenge [123,124,129].

To overcome the intrinsic limitations of the laparoscopic approach, robotic surgery has been advocated to facilitate the lymph node dissection and complex reconstructions after gastrectomy, even without strong evidence [130]. Since the first robot-assisted gastrectomies described by Hashizume and Giulianotti in 2003 [131,132], robotic surgery has been claimed as safe and assures oncological safety in advanced gastric cancer patients [131,133,134]. When compared with laparoscopic subtotal and total gastrectomy, robotic approach results in decreased blood loss (176.6 vs. 212.5, respectively) and a higher number of retrieved lymph nodes (33 vs. 31.3, respectively) [135], especially supra-pancreatic [136]. These findings further support the hypothesis that the robotic wide range of motion allows us to overcome the abovementioned non-compliance lymphadenectomy. In our experience, the mean number of retrieved lymph nodes was 31.9 nodes in D2 dissection and a mean of 2.7 lymph nodes results positive [137]. Eastern retrospective studies highlighted no differences in stage-specific survival analysis between laparoscopic and robotic gastrectomy, in view of long operation time and high costs [136].

The latest published umbrella review on this topic, performed on more than 37,500 subjects, showed no difference in margins infiltration, conversion rate, postoperative compli-

cations, and hospital stay and mortality, between robotic and laparoscopic groups [138]. Interestingly, in 12 meta-analyses, the number of lymph nodes harvested in robotic gastrectomy was higher than in the laparoscopic approach and the distal margin of resection is higher.

A study was conducted on a combined cohort of patients from the United States and China which showed no statistically significant difference in the 5-year overall survival rate between minimally invasive and open approaches. A positive trend as TNM stage increased was noted [139]: long-term survival after minimally invasive gastrectomy tended to be higher in patients with stage III gastric cancer, probably due to lower stress responses, better preserved immune function, and earlier administration of adjuvant chemotherapy [139,140].

Although results appear consistent with previous evidence on the subject, it is more complex to define the role of minimally invasive surgery in advanced gastric cancer after neoadjuvant chemotherapy. Adhesions of tissues, destruction of anatomical dissection planes, perigastric edema and tissue fibrotic changes, occurring after chemotherapy, could increase the surgical difficulty. However, laparoscopy has several advantages, such as delicate manipulation, visual magnification, better exposure, faster recovery, and damage control which might reduce the surgical risk following preoperative chemotherapy.

Investigating the results of quality of life and 3-year overall survival, the European STOMACH trial proved that minimally invasive resection after neoadjuvant chemotherapy is as safe as open total gastrectomy, with a similar number of resected lymph nodes (41.8 vs. 43.4, respectively) and radicality (44 vs. 48 R0 resection, respectively) [141]. However, the inadequate dissection of lymph node station 10 determined a low rate of successful D2 lymphadenectomy. Thus, in the latest JGCA guidelines, standard dissection of station 10 is recommended only if the tumor invades the greater curvature [3].

Previous studies have confirmed that laparoscopic gastrectomy is a feasible method after neoadjuvant therapy, as it was associated with better short- and comparable long-term outcomes (3 years) compared to an open approach [142,143].

As far as we know, only a phase II trial [144] compared laparoscopic and open approach in a neoadjuvant setting [145]. Interestingly, the laparoscopic group had both more adherence to adjuvant chemotherapy (87% versus 70%) and a higher rate of completing therapy (60% versus 40%). Nevertheless, neither the safety and efficacy nor oncologic stability is yet widely established for laparoscopic gastric surgery in a neoadjuvant setting.

In Asian countries, many prospective trials are currently being conducted. The CLASS03a trial [146] and ClinicalTrials.gov NCT02902575 [147] aim to further evaluate safety, feasibility, and long-term outcomes of laparoscopic gastrectomy for advanced gastric cancer after neoadjuvant chemotherapy. In a similar setting, the REALIZATION trial [144] and the ClinicalTrials.gov NCT04658589 [148] will compare open and laparoscopic approach in distal gastrectomy and D2 lymph node dissection.

A further question is whether robotic gastrectomy could offer at least the same benefits as laparoscopic surgery compared to open surgery after neoadjuvant treatment. American data shows on one hand acceptable survival and recurrence rates, on the other hand no difference in R0 and successful lymphadenectomy, while showing an increased node harvest (18 vs. 17 vs. 16 in robotic, laparoscopic, and open gastrectomy, respectively) [99]. Neoadjuvant chemotherapy on the one hand has shown to affect lymph node harvest, on the other hand has proved to be the strongest predictor of open conversion rate, probably because of peritumoral connective tissue changes, liver enlargement, and margin blurring, which definitely led to the loss of visual definition of the proximal border of the tumor [149]. In contrast, neoadjuvant chemotherapy was not a risk factor for common early postoperative complications in Okabe analysis [150].

Because of the paucity of long-term follow-up results, Western reference centers for gastric cancer have implemented prospective data collection, as in the context of UGIRA (The Upper GI International Robotic Association).

In Italy, GIRCG recently investigated laparoscopic and open gastrectomy for advanced gastric cancer, a quarter of whom underwent pre-operative chemotherapy, between 2015

and 2018, showing similar 3-year survival outcomes and 30-day morbidity/mortality [126]. The same group designed a propensity score matching analysis aimed at comparing survival outcomes in patients undergoing curative-intent laparoscopic and robotic gastrectomy for advanced gastric cancer. Results will be presented shortly.

Definite evidence of the safety of minimally invasive surgery for the cure of gastric cancer in the era of neoadjuvant chemotherapy is not available yet; its broader use stressed a shift toward more tailored approach, with the intention of achieving better treatment efficacy for each patient, for each tumor [123,124].

Key points:

- No clear superiority of minimally invasive surgery has been proved.
- Minimally invasive surgery after neoadjuvant chemotherapy is safe and effective if performed in referral centers.
- Further trials are required to establish the viability and long-term outcomes of robotic gastrectomy.

### 4. Ongoing Trials and Future Perspectives

ClinicalTrials.gov NCT03961373: A phase III randomized controlled trial comparing D2 and D2plus lymphadenectomy after neoadjuvant treatment in patients with advanced gastric cancer, without clinically detectable PAN involvement and no evidence of distant metastases (stage IIA-IIIC), has just started in our center in Siena, Italy. In both groups, surgical treatment will be performed within 4–6 weeks after the end of the last cycle of neoadjuvant chemotherapy. The task of the trial "Neo-D2plus" is to assess if neoadjuvant chemotherapy together with more extended lymphadenectomy may improve the outcome of patients. Secondary endpoints are the patients' response to chemotherapy, resection rate, quality of life, complications attributable to surgical intervention, side effects of chemotherapy, and the duration of hospitalization [151].

ClinicalTrials.gov NCT02139605: An Indian phase III randomized controlled trial aims to compare D2 and D3 lymphadenectomy after neoadjuvant treatment (after at least one complete cycle) for non-metastatic, locally advanced but resectable gastric cancer. Primary and secondary outcomes are a 5-year overall survival rate and a 2-year disease-free survival. Previously, just one multi-institutional, non-randomized study compared D2 and D3 lymphadenectomies, and reported that D3 had survival advantages in tumors of 50–100 mm in diameter, with or without metastatic nodes [152].

JCOG1704: The phase II "Bulky/PAN-GC DOS NAC" trial aims to evaluate the efficacy and safety of neoadjuvant chemotherapy with docetaxel, oxaliplatin and S-1 (DOS) followed by D2+ PAN dissection in advanced gastric cancer with ELM. The aim is to evaluate the pathological response rate (defined as patients achieving grade two or more according to the histological criteria of the Japanese Classification of Gastric Carcinoma [17]). Secondary endpoints are the overall survival, and the relapse-free survival, in patients with R0 resection, the proportion of R0 resection, the proportion of completion of surgery, the proportion of completion of protocol treatment, and the response rate of preoperative chemotherapy [153].

### 5. Conclusions

Gastric cancer remains a major global health problem, in terms of incidence and mortality. D2 dissection is the well-established standard treatment for resectable gastric cancer in most of the current guidelines. Despite this statement, the definition of the "optimal" approach for lymphadenectomy in gastric cancer patients is still debated. Surgeons must be trained to perform more extended lymphadenectomy in advanced stages with high risk of metastases to distant nodes. In patients with a node-positive disease, the number of lymph nodes harvested is an independent predictor of survival, and relates to the number of lymph node metastases.

Current guidelines recommend preoperative chemotherapy for resectable gastric cancer with clinical stage $\geq$ T2 or N+. Patients with clinically positive para-aortic nodal

metastases should undergo neoadjuvant chemotherapy followed by therapeutic PAND. In any case, prophylactic PAND and posterior nodes dissection after neoadjuvant treatment may be performed in patients at high risk of PAN metastases, such as positive bulky nodes in the second level perigastric nodal stations or diffuse histotype. GIRCG started a randomized controlled trial to explore the role of prophylactic D2+ compared to the standard D2 in locally advanced gastric cancer patients treated with perioperative chemotherapy in the Western countries.

To paraphrase Oscar Wilde, "*nowadays people know the price of everything and the value of nothing*"; modern surgeons should balance possible oncological value of D2plus lymphadenectomy with the price of post-operative complications and the risk of mortality. Research is developing fast and progress in health care delivery and new insights into tumor burden allow for more precise classifications, more efficacious therapies, more efficient tailored surgery, and better quality of care.

**Author Contributions:** Conceptualization, L.M., L.C. and D.M.; methodology, L.C. and G.E.P.; validation, L.V. and S.A.P.; investigation, L.M. and L.C.; data curation, L.C. and D.M.; writing—original draft preparation, L.C.; writing—review and editing, L.C., G.E.P. and V.R.; visualization, S.A.P. and V.R.; supervision, F.R.; project administration, F.R. and D.M. All authors have read and agreed to the published version of the manuscript.

**Funding:** This research received no external funding.

**Conflicts of Interest:** The authors declare no conflict of interest.

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
