# Peer review of "Extended Lymphadenectomy for Gastric Cancer in the Neoadjuvant Era: Current Status, Clinical Implications and Contentious Issues"

_curroncol, doi:10.3390/curroncol30010067_

Round 1
Reviewer 1 Report
The article is interesting even if some observations are mandatory.
1- Since the conclusions of Sasako (2006) and Ri (2021), the statement regarding the prophilactic PAND should be presented as a personal point of view
2- Too many self-citations (27) are present in References; it would be advisable to reduce their number, limiting to the more recent ones, in my opinion
3- References 43, 64 and 96 are the same publication
4- A language polishing should be done
Author Response
Dear Reviewer,
grateful for your comments.
- We changed the sentence.
- We reduced them. However, our manuscript aims to be a comprehensive review of the literature and analyzes almost all the articles on the topic.
- The references cite the same book, but different chapters. We detailed them.
- We provided language editing.
Reviewer 2 Report
The manuscript is well-written and readable; it contributes significantly to this field, collecting the most appropriate and recent references.
I have no comment for the authors.
Author Response
Dear Reviewer, we really thank for your consideration. Best regards.
Reviewer 3 Report
The authors showed the value of extended lymphadenectomy for gastric cancer. They described current adaptation of D2, D2 plus and D3 lymphadenectomy, and benefits of the collection of more than twenty-five lymph nodes and para-aortic nodes dissection after neoadjuvant chemotherapy. Although this manuscript has been well-written, some minor issues remain to be revised.
1. The title of section “1.1 D2 lymphadenectomy D2 plus and D3 lymphadenectomies” (at the line 26) might be “1.1 D2 lymphadenectomy”.
2. In the section, subheadings, 1. (at the line 45), 2. (at the line 69), 3. (at the line 107), and 4. (at the line 109), may correspond to the four clinical benefits, (1)-(4) (at the lines 40-44). It is a little confusing. The short title should be described in the subheadings. In addition, the lines 297 and 306 were same as the section 1.1.
3. In the section 3.1, the authors described molecular characteristics of gastric cancers (at the lines 306-322). They showed clinical trials of microsatellite instability (MSI) and microsatellite stable (MSS) gastric cancers with perioperative or adjuvant chemotherapy. However, it may be still controversial. Some clinical trials included only a small number of patients with MSI-high gastric cancer. They should comment about it. They had also better comment about recent immunotherapy/ immunochemotherapy for MSI-high gastric cancers. In addition, The Cancer Genome Atlas (TCGA) proposed four molecular subtypes of gastric cancers, e.g., EB virus-positive gastric cancer. They should comment about other molecular subtypes.
Author Response
Dear Reviewer,
grateful for your insights.
- Done.
- We corrected.
- In the section 3.1 we speculated that despite MSI patients should have a better prognosis compared with MSS gastric cancers, they may not benefit from chemotherapy and more extended lymphadenectomy. However, they may benefice from immunotherapy. We agree that some clinical trials included small cohorts of MSI-H patients. We commented it.
According to TCGA project, we analyzed (1) MSI and (2) GS patients. The latter overlapped with MSS/EMT group in ACRG classification, sharing similar molecular mutations (i.e. CDH1). (3) CIN tumors have predominantly mutations in ErbB2. It is known that trastuzumab can be considered in HER2 mutation, whereas TOGA trial failed to demonstrate a considerable improved survival (only 2 months). For this reason, we described only new trials in line 316-325. Finally, (4) EBV has been now described.